# COVID-19 as an Occupational Disease—Temporal Trends in the Number and Severity of Claims in Germany

**DOI:** 10.3390/ijerph20021182

**Published:** 2023-01-09

**Authors:** Albert Nienhaus, Johanna Stranzinger, Agnessa Kozak

**Affiliations:** 1Competence Center for Epidemiology and Health Services Research for Healthcare Professionals (CVcare), Institute for Health Services Research in Dermatology and Nursing (IVDP), University Medical Center Hamburg-Eppendorf (UKE), 20246 Hamburg, Germany; 2Department for Occupational Medicine, Hazardous Substances and Health Sciences (AGG), Statutory Accident Insurance and Prevention in the Health and Welfare Services (BGW), 22089 Hamburg, Germany

**Keywords:** COVID-19, occupational health, occupational disease, vaccination

## Abstract

COVID-19 is considered an occupational disease (OD), when infection occurs at the workplace for health workers (HW). Because of the increased infection risk of these workers, they were deemed to be a priority group when the vaccination campaign started in Germany in December 2020. By December 2021, more than 90% of HW had been vaccinated twice. We studied the number and the time trend concerning the severity of OD claims related to COVID-19. Workers’ compensation claims for OD are recorded in a standardized database of the Statutory Accident Insurance and Prevention in the Health and Welfare Services (BGW). We analyzed all notifiable COVID-19 related claims filed between 1 March 2020 and 30 September 2022. The proportion of severe cases was estimated by inpatient stays, injury benefit payments, rehabilitation measures, and deaths. The data analysis was descriptive. Due to COVID-19, 317,403 notifiable cases were reported to the BGW. Of these, 200,505 (63.2%) had thus far been recognized as OD. The number of notifiable cases was highest in 2022 and lowest in 2020. In total, 3289 insured individuals were admitted to rehabilitation management. This represented 1.6% of all recognized ODs due to COVID-19 at the BGW. The proportion of cases admitted to rehabilitation management decreased from 4.5% of all recognized ODs in 2020 to 3.2% in 2021 and to 0.1% of all recognized cases in 2022. For inpatient stays, injury benefit payment, and death, a similar trend was observed. Therefore, it might be concluded that the successful vaccination campaign mitigated the negative health effects of COVID-19 on HW. Even with vaccination, severe cases can occur. Therefore, infection prevention at the workplace remains paramount.

## 1. Introduction

Healthcare workers (HW) have a higher occupational risk of being infected and becoming ill with the SARS-CoV-2 pathogen [1,2,3]. The risk of infection was increased when personal protective equipment (PPE) was in short supply [4]. In the USA, out of all claims of an occupational disease (OD) due to COVID-19, 73% concerned HW [5]. In a German cross-sectional study after the first COVID-19 wave, the number and duration of contacts with COVID-19 patients were risk factors for a SARS-CoV-2 infection [6]. In a German longitudinal study covering the period June 2020 to May 2021, working in intensive care was associated with a fourfold increase in the risk of infection, and working in other wards was associated with a twofold increase in the risk of infection, compared to hospital personnel without regular contact with patients [7]. HW were a prioritized group for their own protection and that of patients, as the vaccination campaign began in Germany on 27 December 2020. The vaccines used were effective in protecting against symptomatic infections and severe courses of disease in the first few months after the double vaccination and, beyond that, after booster vaccinations [8,9,10]. A register-based cohort study from Finland comprising HW showed that protection from hospitalization due to COVID-19 ranged from 98% to 100% depending on the vaccination schedule during the period in which the Delta variant of the coronaviruses predominated [11]. According to data from the USA, in the Omicron wave (January to April 2022), the hospitalization rate was 10.5 times higher among unvaccinated individuals and 2.5 times higher among individuals without booster vaccination than among those who had received all three vaccine doses [12].

In Germany, the willingness of HW to be vaccinated appeared to be rather low before the vaccination campaign [13,14]. The nursing staff were especially skeptical. At the time, their willingness to be vaccinated was only 50%. However, after vaccines became available and the vaccination campaign commenced on 27 December 2020, vaccination readiness seems to have increased. Our own online survey conducted in March and April 2021 found that 62% had already been vaccinated, and a further 22% were willing to be vaccinated. Only 9% refused vaccination at this time [15]. According to a survey by the Robert Koch Institute (RKI) in April 2022, 96% of HW had been vaccinated [16]. To this extent, the vaccination campaign can be said to have been successful.

In Germany, COVID-19 is considered an OD, which can be compensated by mandatory social insurance for work-related accidents and disease, whenever it is likely that the infection occurred at the workplace of health, social, or laboratory workers [17]. By May 2021, more than 50,000 ODs due to COVID-19 were already recognized in health and social workers by the BGW, the compensation board of private health and social work providers [18]. Hospitalization rates for these workers was 0.4%, and the mortality rate was 0.09%. Despite the vaccination campaign, in 2021 and 2022, the number of SARS-CoV-2 infections increased in the general population in Germany, which was attributed to the different variations of SARS-CoV-2 and changes in the general prevention concepts. This increasing number of infections in the general population gave rise to work-related infection risk. However, it might also explain why, in a study using job titles, the relative risk for mortality in HW decreased from the beginning of the pandemic to the year 2022 [19]. However, a study from Switzerland showed that HW had an increased infection risk after the Omicron variant become predominant [20].

Therefore, we investigated the number of ODs in HW in the first, second, and third year of the pandemic in Germany. The time trends of the severity of the course of COVID-19 among HW before, after, and especially during the year of the vaccination campaign were analyzed.

## 2. Materials and Methods

COVID-19 is recognized as an OD under No. 3101 of the Occupational Diseases Ordinance for workers in the healthcare sector, in welfare, or in laboratories and occupations with a similar increased risk of infection. Since March 2020, all reports made to the BGW concerning SARS-CoV-2 or COVID-19 have been systematically registered in a separate documentation system [21]. Cases are notifiable when the virus is identified by PCR, the infection is symptomatic, and when it is suspected that infection occurred at the workplace. Injury benefits are paid after the claim is accepted as OD, and the sick leave due to the OD is longer than six weeks. The documentation system records test results for SARS-CoV-2, inpatient stays, injury benefit payments, payments for medical treatment, inclusion in rehabilitation management, and deaths. In order to estimate the severity of COVID-19, the proportion of hospitalized cases and cases with payment of injury benefits or for medical services, as well as deaths from notifiable or recognized OD, were separately analyzed by year. For the year 2021, an evaluation was also carried out by calendar week in order to estimate the effect of the increasing vaccination rate over the course of the year. The analysis is based on the BGW’s special assessment of COVID-19 cases from March 2020 to the end of September 2022.

The data analysis was carried out descriptively. No information on the vaccination status of the HWs with a COVID-19 related claim was available. Therefore, an ecological study was performed.

The data were evaluated anonymously in aggregated form and presented descriptively. All data protection regulations were fulfilled. An ethics vote was not obtained, as anonymous secondary data were used for the analysis.

## 3. Results

As of September 2022, 317,403 notifiable cases due to COVID-19 had been reported to the BGW. Of these, 200,505 (63.2%) had thus far been recognized as OD (Table 1). The number of notifiable cases was highest in the year 2022 and lowest in the year 2020. For the year 2020, the recognition rate was 78.2%. In total, the course of COVID-19 has so far been severe enough for 3289 insured individuals to be admitted to rehabilitation management. This represented 1.6% of all recognized occupational diseases due to COVID-19 at the BGW. The proportion of cases admitted to rehabilitation management decreased from 4.5% in 2020 to 3.2% in 2021 and to 0.1% in 2022. The proportion of cases that received benefits for medical treatment decreased from 13.3% in 2020 to 1.6% in the first three quarters of 2022. So far, medical treatment benefits have been provided for 16,434 insured individuals. This corresponds to 5.2% of all notifiable cases.

Inpatient treatment for COVID-19 has been needed for 3944 insured individuals of the BGW so far (Table 2). This corresponded to 1.2% of all notifiable cases. Once again, there was a significant decrease in the proportion of those affected in terms of the number of reports per year: from 4.5% in 2020 to 0.04% in 2022. Injury benefits due to COVID-19 were received by 5077 insured individuals (1.6% of all reported cases) who were unable to work for longer than six weeks. The proportion of the reports was 4.2% in 2020 and 0.1% in 2022. This was a reduction of 97.6%. For the 162 deaths (0.05% of all notifications), in total, the proportion in the first three quarters of 2022 was about one-tenth of the proportion of cases in 2020 (0.15% versus 0.014%).

Between March 2020 and September 2022, three waves of claim notifications were distinguished (Figure 1). The first wave reached a maximum with about 9500 cases in week 7 of 2020. The second wave started to increase in week 44 (October) in 2020 and reached a maximum of close to 5500 cases in week 7 (February) of 2021. The third wave started to increase in week 47 (November) of 2021 and reached its maximum of more than 8000 cases in week 19 (May) of 2022. Therefore, the reported cases from 2021 were not evenly distributed throughout the year. In the first months of 2021 (week one to four), the number of reported cases was the highest. During the summer months, the number of reports was low, and it increased again in autumn.

Figure 2 shows the proportion of cases (%) needing inpatient treatment among the notifiable cases in the respective calendar weeks of 2021. Apart from the three outliers in August and September (weeks 31, 34, and 36), there was a continuous decrease in the proportion of inpatient treatment cases from roughly 3.5% in the first four weeks to less than 0.5% in the last four weeks of 2021.

The proportion of cases that received injury benefits due to COVID-19 decreased significantly (Figure 3). However, this decline was most pronounced in the last four months of 2021. With regard to the number of deaths, no trend could be identified throughout the year 2021. The proportion here fluctuated between 0 and 0.37% with no obvious pattern (data not shown).

## 4. Discussion

COVID-19 has entirely changed the occupational occurrence of infections in Germany. In the years before the pandemic, roughly 800 to 1000 claims of infectious diseases subject to mandatory reporting were submitted to the BGW each year [22]. In a previous publication covering the first four months of the pandemic, the BGW had already received 4398 claims due to SARS-CoV-2 and COVID-19, which were subject to mandatory reporting. Eleven deaths and 151 severe illnesses needing hospitalization demonstrated the particular vulnerability of healthcare workers [21]. These numbers increased to 84,728 claims, 375 hospitalizations, and 77 deaths until May 2021 [18]. And the numbers further increased during the next months (see below). This means the number of claims increased more than the number of deaths (19 times versus seven times for deaths and 2.5 times for hospitalizations). The steep increase in claims is well explained by the first three pandemic waves in Germany. The first wave at the beginning of the pandemic was less severe than the second wave starting in October 2020 and the third wave starting in March 2021. The first wave had a maximum of 5000 new cases per day, and the second and third waves had a maximum of 25,000 and 20,000 per day, respectively. However, the difference in the increase in claims compared to the hospitalizations and deaths might indicate that more cases with a lighter case of the disease were reported. Several factors might explain this development. There might have been more non diagnosed infections at the beginning [23]. In addition, at the beginning of the pandemic, it was not clear under which terms COVID-19 would be accepted as an OD. This became clearer with a publication explaining the conditions under which COVID-19 was recognized as an OD [17]. Probably more importantly, only during the course of 2020 did it became evident that long-lasting illness could occur even after a light case of the acute infection [24]. Therefore, milder forms of the infection might have been reported to make sure that the infection was registered in case symptoms occur later.

Although the number of notifiable cases increased from the year 2020 to the first nine months of 2022, we observed a significant decrease in the proportion of cases with a severe course as measured by the various indicators for the severity of COVID-19 among health and social workers in Germany. There was a risk reduction by a factor of 10 or more in some cases. This trend could be seen in the proportion of hospitalization, injury benefits, and rehabilitation management cases, as well as in the number of cases involving medical benefits and deaths. We expected the proportion of severe courses of COVID-19 to decrease in terms of the cases reported per calendar week as the vaccination campaign progressed in 2021. According to the data presented here, this association was confirmed for the proportion of inpatient treatment cases in 2021. This saw a linear decrease over the calendar weeks from initially around 3.5% to less than 0.5%. There was also a decreasing trend in the proportion of cases involving benefits for medical treatment, but this was not linear. Here, the decrease was particularly evident in the last quarter of 2021. There was no temporal trend in the proportion of deaths over the course of 2021. In this regard, the proportion was only reduced in comparison to 2020 (0.15% versus 0.095%). It is possible that no temporal trend could be identified for the year 2021 due to the small overall number of deaths (*n* = 105 distributed over 52 weeks), which led to random variation.

We do not have any information regarding vaccination for the notifiable COVID-19 cases. However, the number of reports continued to increase despite vaccination. The increased number of reports also correlated to the increased number of infections in the German general population [24]. On the one hand, these were caused by the increasing relaxation of contact bans during the pandemic and, on the other hand, by a change in the dominant virus variants. At the end of 2020 and during the first half of 2021, the Delta variant predominated in Germany. Starting in the fourth quarter of 2021, this variant was displaced by the Omicron variant. The Omicron variant has been dominant in Germany since December 2021 and continues to be so in its various forms. A clear example is the celebration of carnival in the year 2021 and 2022 in Germany. In 2021, no mass meetings were allowed, and in 2022, carnival was celebrated in the streets with no restrictions. In 2021, no increase in the COVID-19 incidence was observed, while in 2022, a sharp increase occurred two weeks after the carnival days in Cologne, one of the centers celebrating carnival in Germany [25]. The Omicron variant is more infectious but less lethal than the Delta variant. This explains why the number of infections has increased, despite the vaccination of both the general population and prioritized groups such as healthcare and social workers. The variant change may explain a decline in the severity of COVID-19 courses for the final weeks of 2021 but not the decline we already observed for the proportion of inpatient treatments in the second quarter of 2021. This decline was probably attributed to vaccination. The effect was then amplified by the change to a variant with higher infectivity but less frequent severe courses. This interpretation was supported by other studies showing that the vaccines reduced not only the transmission but also the severity of disease in infected workers during the Delta phase in the first few months after primary immunization [10,11].

This is an ecological study; therefore, as a limitation, causal interpretations can only be made with caution. We know that the estimates for vaccination uptake in German health workers increased during 2021. However, it is not known whether a claimant was vaccinated or not. Therefore, conclusions at the individual level are not possible. The virus variant that infected the insured individuals was not known. It was therefore not possible to assess what influence it had on the probability of a severe course. A change in reporting behavior over time may also have occurred. The discussions regarding long COVID even after mild courses may have led to an increased number of mild cases being reported, e.g., in collective reports from social health insurances or employers. In addition to reporting behavior, there was also a time lag in processing cases and costing. This effect may apply to cases reported in 2022, but this lag cannot explain the downward trend in case severity in 2021. In light of the extensive literature on the protective effects of COVID-19 vaccination and infection or both [26,27,28,29], we believe it is likely that the observed association is likely to indicate a causal relationship.

Infection after complete vaccination and reinfection after being vaccinated is possible. These infections might cause severe consequences of the disease, as was shown by data from the U.S. veteran administration [30]. Therefore, even after a successful vaccination campaign, infection protection at the workplace remains paramount.

## 5. Conclusions

Although the vaccination status of the BGW-insured individuals was not known, the BGW data indicated that the severity of COVID-19 cases consistently decreased as vaccination coverage increased during the Delta period. The proportion of the reported insured events that did not need any insurance benefit payment also increased accordingly. This raises the question as to whether reporting “almost” asymptomatic SARS-CoV-2 infections and mild COVID-19 courses still makes sense, given the large administrative workload for processing these files. It is possible that we have now entered the endemic phase of COVID-19 and that SARS-CoV-2 infections should be treated in a similar way to influenza infections for HW: yes to vaccination but only report an infection in cases where a benefit for the claimant is expected. With this paper, we hope to stimulate a discussion concerning the administration and compensation of HW between different countries.

## Figures and Tables

**Figure 1 ijerph-20-01182-f001:**
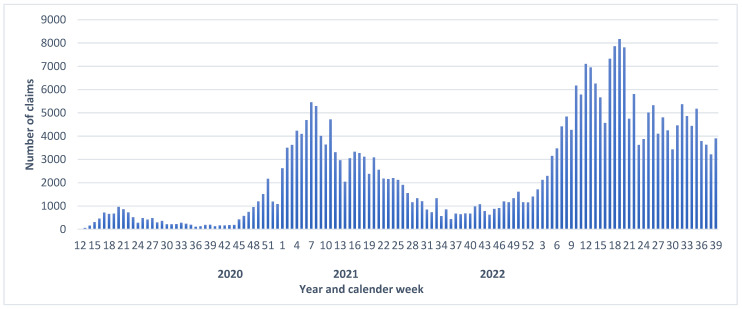
Number of notifiable cases due to COVID-19 from March 2020 to September 2022, separated by calendar week of the notification.

**Figure 2 ijerph-20-01182-f002:**
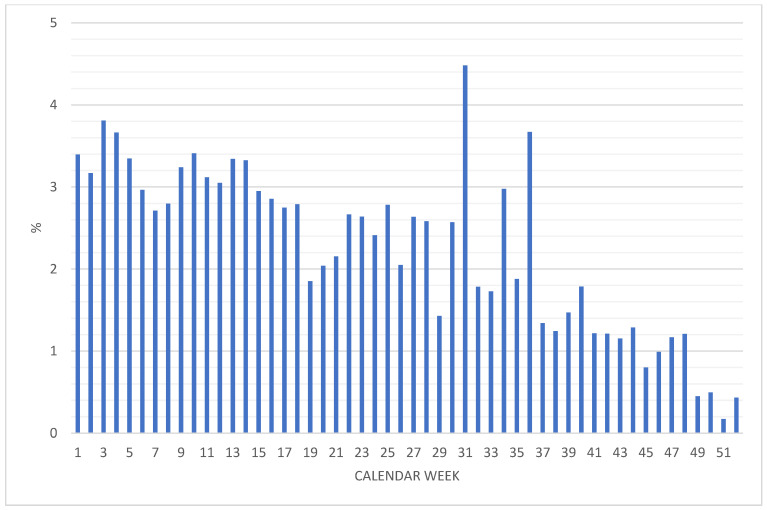
Proportion of cases (%) involving inpatient treatment among the reported cases in the respective calendar weeks of 2021.

**Figure 3 ijerph-20-01182-f003:**
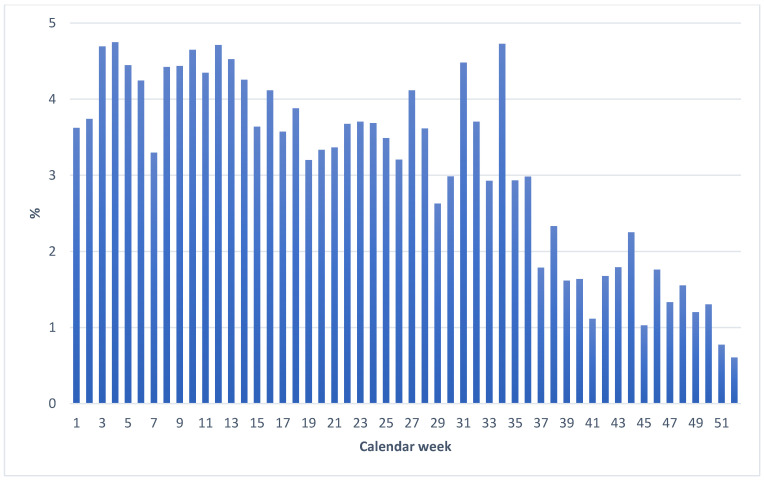
Proportion of cases (%) involving injury benefit among the reported cases in the respective calendar weeks of 2021.

**Table 1 ijerph-20-01182-t001:** Notifiable and recognized occupational diseases due to COVID-19, as well as the number of cases in rehabilitation management and cases with medical treatment, separated by year of reporting.

Year	Notifiable Cases	Recognized Cases		Rehabilitation Management		Benefits for Medical Treatment	
	*N*	*n*	%	*n* *	%	*n* **	%
2020	21,147	16,538	78.2	747	4.5	2816	13.3
2021	111,099	76,164	68.6	2420	3.2	10,752	9.7
September 2022	185,157	107,803	58.2	122	0.1	2866	1.6
Total	317,403	200,505	63.2	3289	1.6	16,434	5.2

* % of notifiable cases, ** % of reported cases.

**Table 2 ijerph-20-01182-t002:** Number and proportion of cases with inpatient treatment, receipt of injury benefit, or deaths due to COVID-19 among insured individuals of the BGW with a notifiable occupational disease.

Year	Inpatient Treatment		Injury Benefit		Death	
	*N*	%	*N*	%	*n*	%
2020	947	4.5	884	4.2	31	0.15
2021	2914	2.6	4012	3.6	105	0.095
September 2022	83	0.04	181	0.1	26	0.014
Total	3944	1.2	5077	1.6	162	0.05
Reduction from 2020 to 2022		99.2		97.6		90.7

## Data Availability

Data are available upon request. Please send email to: albert.nienhaus@bgw-online.de.

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
