# Peer review of "COVID-19 as an Occupational Disease—Temporal Trends in the Number and Severity of Claims in Germany"

_ijerph, 2023, doi:10.3390/ijerph20021182_

Round 1
Reviewer 1 Report
This study shows interesting data on medical certification of occupational COVID-19 in Germany. Only the presentation of data in figures no 2, 3, 4 is not entirely convincing to me, although they are understandable. A valuable conclusion is the proposal of a top-down reduction of bureaucracy in the case of clinically asymptomatic infections. Similiar studies from other countries are needed!
Author Response
Reviewer 1
- This study shows interesting data on medical certification of occupational COVID-19 in Germany. Only the presentation of data in figures no 2, 3, 4 is not entirely convincing to me, although they are understandable.
Response: We agree with the reviewer. It would be nicer to present the data by month instead of calendar weeks. However, as we had aggregated data only, it is not possible to switch from calendar week to month. We noticed that we had the Germany style for presenting numbers. This might have been confusing. We changed it.
- A valuable conclusion is the proposal of a top-down reduction of bureaucracy in the case of clinically asymptomatic infections. Similiar studies from other countries are needed!
Response: we are grateful for this comment. The conclusion was added.
Reviewer 2 Report
The paper reports interesting data on occupational COVID-19 in Germany. The paper is valuable and deserve publication. I suggest to improve the paper in the following ways
1. The missing data about vaccination is a limitation of the study that need to be discussed more deeply
2. I suggest to insert trend of infection for the overall period considered and not only for the 2021. I think that will be more informative
3. Abstract “ We therefore studied the association between vaccination uptake and the 14 number and severity of OD claims related to COVID-19. Workers’ compensation claims for OD are recorded in a standardized database of the 15 Statutory Accident Insurance and Prevention in the Health and Welfare Services (BGW).”
I think that it is very difficult to say that, considering that data on vaccination status was not available. In the paper the association with vaccination uptake and number and severity of OD claims is impossible, because data on vaccination is not available.
I suggest to change the sentence considering only available data.
4. Title. I suggest to delete the last sentence because no data was available on vaccination status
Minor
- Please report in English the name of Months (May instead of Mai etc)
Author Response
Reviewer 2
The paper reports interesting data on occupational COVID-19 in Germany. The paper is valuable and deserve publication. I suggest to improve the paper in the following ways
- The missing data about vaccination is a limitation of the study that need to be discussed more deeply
Response: We agree. As we already mentioned we conducted an ecologic study. Now we discuss in more detail the shortcomings of ecologic studies. In addition we down tuned the conclusion (see comment to academic editor) Now we write in the limitation part:
This is an ecological study; therefore, as a limitation causal interpretations can only be made with caution. We know that the estimates for vaccination uptake in German health workers increased during 2021. However, it is not known, whether a claimant was vaccinated or not. Therefor conclusions on the individual level are not possible.
- I suggest to insert trend of infection for the overall period considered and not only for the 2021. I think that will be more informative
Response: Thank you for this comment. We changed Figure 1. It covers the whole period of the pandemic until September 2022. Three waves can be distinguished.
- Abstract “ We therefore studied the association between vaccination uptake and the 14 number and severity of OD claims related to COVID-19. Workers’ compensation claims for OD are recorded in a standardized database of the 15 Statutory Accident Insurance and Prevention in the Health and Welfare Services (BGW).”
I think that it is very difficult to say that, considering that data on vaccination status was not available. In the paper the association with vaccination uptake and number and severity of OD claims is impossible, because data on vaccination is not available.
I suggest to change the sentence considering only available data.
Response:
We changed the sentence accordingly. Now we write: We studied the number and the time trend concerning severity of OD claims related to COVID-19. Workers’ compensation claims for OD are recorded in a standardized database of the Statutory Accident Insurance and Prevention in the Health and Welfare Services (BGW).”
- Title. I suggest to delete the last sentence because no data was available on vaccination status
Response: We deleted the last sentence. By that and by rephrasing the objective we focus more on the description of the claims and the time trend observed concerning the severity of the disease.
Minor
- Please, report in English the name of Months (May instead of Mai etc)
Response: Thank you for pointing this out. We changed the figure with the mistake.
Thank you for the thoughtful comments of the editor and the reviewer. We think they helped to further improve the manuscript.
Round 2
Reviewer 2 Report
The paper is now suitable for publication